# Mirroring Skyrmions in Synthetic Antiferromagnets via Modular Design

**DOI:** 10.3390/nano13050859

**Published:** 2023-02-25

**Authors:** Panluo Deng, Fengjun Zhuo, Hang Li, Zhenxiang Cheng

**Affiliations:** 1School of Physics and Electronics, Henan University, Kaifeng 475004, China; 2School of Physics Science and Technology, ShanghaiTech University, Shanghai 201210, China; 3Institute for Superconducting and Electronic Materials, Australian Institute of Innovative Materials, Innovation Campus, University of Wollongong, Squires Way, North Wollongong, NSW 2500, Australia

**Keywords:** magnetic skyrmion, interlayer exchange coupling, synthetic antiferromagnets, micromagnetic simulation

## Abstract

Skyrmions are promising for the next generation of spintronic devices, which involves the production and transfer of skyrmions. The creation of skyrmions can be realized by a magnetic field, electric field, or electric current while the controllable transfer of skyrmions is hindered by the skyrmion Hall effect. Here, we propose utilizing the interlayer exchange coupling induced by the Ruderman–Kittel–Kasuya–Yoshida interactions to create skyrmions through hybrid ferromagnet/synthetic antiferromagnet structures. An initial skyrmion in ferromagnetic regions could create a mirroring skyrmion with an opposite topological charge in antiferromagnetic regions driven by the current. Furthermore, the created skyrmions could be transferred in synthetic antiferromagnets without deviations away from the main trajectories due to the suppression of the skyrmion Hall effect in comparison to the transfer of the skyrmion in ferromagnets. The interlayer exchange coupling can be tuned, and the mirrored skyrmions can be separated when they reach the desired locations. Using this approach, the antiferromagnetic coupled skyrmions can be repeatedly created in hybrid ferromagnet/synthetic antiferromagnet structures. Our work not only supplies a highly efficient approach to create isolated skyrmions and correct the errors in the process of skyrmion transport, but also paves the way to a vital information writing technique based on the motion of skyrmions for skyrmion-based data storage and logic devices.

## 1. Introduction

The magnetic skyrmion is a swirling spin texture that is topologically stable [1,2,3,4,5,6,7,8]. It has attracted considerable interest due to its promising role in next-generation spintronic devices [9,10,11,12,13,14]. The formation of magnetic skyrmions is driven by competition between the Dzyaloshinskii–Moriya interaction (DMI) [15,16,17], the magnetic exchange interaction, and magnetocrystalline anisotropy. Advances in materials research have led to the discovery of magnetic skyrmions in metallic multilayers [18,19,20,21], multiferroic magnets [22], Heusler compounds [23], perovskites [24], and two-dimensional van der Waals (vdW) materials [25,26].

Recent reports on skyrmions in synthetic antiferromagnets (SAFs) have caught our attention [27,28,29]. The absence of the skyrmion Hall effect, in particular, is a tantalizing feature [30,31,32,33]. A SAF is a multilayer heterostructure that consists of two oppositely magnetized ferromagnetic layers separated by a normal spacer, which emulates the spin configuration of an A-type antiferromagnet (AFM) [34]. The interlayer coupling in the SAF is mediated by itinerant electrons [35,36,37,38,39] and is largely dependent on the electronic properties of the normal spacer [40,41,42,43,44]. Therefore, the SAF usually has a lower (than intrinsic AFM) interlayer exchange coupling, making it more susceptible to DMI and intralayer interactions, and hence a broader range of spin textures. Magnetic skyrmions in SAFs have also exhibited better thermal stability [27,28] and higher speed than in ferromagnetic hosts [30,45]. Recently, creating isolated skyrmions in the SAF through a confined geometry has been reported [46,47]. In fact, the isolated skyrmion could also be created by interlayer exchange coupling, which may be benefit the create of skyrmions in batches.

In a multilayer system, the interlayer exchange coupling also controls the magnetic properties. For example, by altering the layer stacking style and thus the interlayer exchange coupling, we may expand the parametric ranges of skyrmion stabilization and consequently modify the spin configurations. In fact, the effective field due to interlayer exchange coupling has a maximum magnitude ranging from 100 to 200 mT, which is sufficient to tilt spins. In the presence of a current in one ferromagnetic sublayer, the interlayer exchange coupling mediates, via itinerant electrons, the transfer of spin-angular momentum from one sublayer to the other, stabilizing local spins in the neighboring sublayer. In an SAF with skyrmions, this mechanism may lead to a unique mirroring effect: with a sufficiently large spin current, a stable skyrmion in one ferromagnetic sublayer may introduce a skyrmion in the neighboring ferromagnetic layer via the exchange interaction in the vicinity of the skyrmion core. More interestingly, since the skyrmion moves in a pair with its “mirroring image” (in the other layer), the skyrmion Hall effect is largely suppressed due to the cancellation of the Magnus force.

In this work, we propose a hybrid tandem structure, as shown in Figure 1a, to study such a skyrmion mirroring effect. In addition to the multilayers along the thickness (the *z* direction), we introduced a steplike inhomogeneity along the direction of the current flow (the *x* direction): One section (right) of the structure is essentially one ferromagnetic (FM) multilayer, and the other (left) section is an SAF consisting of two ferromagnetic (FM) sublayers. When a single skyrmion moves from the FM section into the SAF, a skyrmion with mirroring spin configuration is created in the other sublayer of the SAF. This transition, namely, from a single ferromagnetic skyrmion to a pair of antiferromagnetically coupled skyrmions, is attributed to the spin-transfer torque mediated mainly by the Ruderman–Kittel–Kasuya–Yoshida (RKKY) interaction, as depicted in Figure 1b.

## 2. Computational Details

### 2.1. Steplike Tandem Structure

To realize the transition between an isolated skyrmion and a pair of antiferromagnetic coupled ones, we proposed a steplike tandem structure, as shown in Figure 1a. For a clear discussion, we considered a homogeneous multilayered tandem structure as one unified “building block” and referred to it as a ferromagnetic module. Figure 1b is a schematic view of the two main ferromagnetic modules in the system. The large bottom module is the base module to generate skyrmions. The SAF section, created by the deposition of another ferromagnetic module on the (left part of) base, provides the inter-modular antiferromagnetic exchange coupling (*A*_inter_) necessary to achieve the mirroring effect. Here, inter-module coupling refers to the coupling between spins in two ferromagnetic modules; couplings within a ferromagnetic module are considered intra-modular.

### 2.2. Methods

Within the micromagnetism framework, the skyrmion motion within each sublayer *i* (*i* = 1 for the top layer and *i* = 2 for the bottom layer) is governed by the Landau–Lifshitz–Gilbert (LLG) equation [48],
(1)$$\frac{dm_{i}}{dt}=-\gamma \mu_{0}m_{i}\times H_{i}^{eff}+\alpha \left[m_{i}\times \frac{dm_{i}}{dt}\right]+\tau_{i}$$
where *γ* is the gyromagnetic ratio; *α* is the Gilbert damping constant; *τ* is the spin torque term; and $H_{i}^{eff}=-(1/\mu_{0}M_{s})\partial \varepsilon_{i}/\partial m_{i}$ is the effective magnetic field, where ε*_i_* is the magnetic energy density and $M_{s}$ is the saturation magnetization. The effective fields include the four contributions including ferromagnetic exchange, magnetic anisotropy, and the demagnetizing field as well as the interfacial DMI [49,50],
(2)$$H_{eff,1}=h_{ex,1}+h_{ani,1}+h_{demag,1}+h_{dmi,1},\;H_{eff,2}=h_{ex,2}+h_{ani,2}+h_{demag,2}+h_{dmi,2}.$$

The exchange fields involve both ferromagnetic coupling between neighbors in each sublayer and the antiferromagnetic coupling between the two sublayers,
(3)$$h_{ex,1}=\frac{{2A}_{intra,1}}{\mu_{0}{M_{s}}^{2}}\nabla^{2}m_{1}+\frac{{4A}_{inter}}{\mu_{0}{{a^{2}M}_{s}}^{2}}m_{2},\;h_{ex,2}=\frac{{2A}_{intra,2}}{\mu_{0}{M_{s}}^{2}}\nabla^{2}m_{2}+\frac{{4A}_{inter}}{\mu_{0}{{a^{2}M}_{s}}^{2}}m_{1}.$$
where $A_{intra,1}$, $A_{intra,2}$ (>0), and $A_{inter}$ (<0) are the intra-sublayer ferromagnetic and inter-sublayer antiferromagnetic exchange constants, respectively; *a* is the lattice constant and $\mu_{0}$ is the permeability of the vacuum. The magnetic anisotropy fields for each sublayer are expressed as:(4)$$h_{ani,1}=\frac{{2K}_{u1}}{\mu_{0}{M_{s}}^{2}}m_{1}\cdot \hat{z},\;h_{ani,2}=\frac{{2K}_{u2}}{\mu_{0}{M_{s}}^{2}}m_{2}\cdot \hat{z}.$$
where $K_{u1}$ and $K_{u2}$ are the uniaxial magnetic anisotropy constants. The demagnetization field is expressed by:(5)$$h_{demag,1}=K_{ij}m_{1},\;h_{demag,2}=K_{ij}m_{2},$$
where $K_{ij}$ is the demagnetizing tensor. For a symmetric multi-layered system, the spin transfer torque (STT) exists in the structure. The Néel skyrmion is driven by an in-plane current applied to the system. In the case of the spin-transfer torque, the Zhang–Li torque term is expressed by [51]:(6)$$\tau_{}^{ZL}=\left[(1+\beta \alpha )m_{i}{\times (m}_{i}\times \left(u\cdot \nabla \right)m_{i})+(\beta -\alpha )m_{i}\times \left(u\cdot \nabla \right)m_{i}\right],\;u=\frac{\mu_{B}\mu_{0}}{2e\gamma B_{sat}\left(1+\beta^{2}\right)}J,$$
where *J* is the current density; *β* is the strength of the non-adiabatic torque; $\mu_{B}$ is the Bohr magneton; and $B_{sat}$ is the saturation magnetization expressed in Tesla. When a skyrmion is driven by a perpendicular spin polarized current, the spin orbit torque (SOT) term is expressed by:(7)$$\tau_{}^{S}=\gamma u\left(m_{i}\times \left(p\times m_{i}\right)\right)-\gamma u^{\prime }\left(m_{i}\times p\right),$$
where $u=\frac{\hbar }{e}\frac{\left|P\right|}{dM_{s}}j_{z}$ is the spin torque coefficient; u’ is the field-like component of the STT coefficient; p is the spin polarization; p = 1 is the spin polarization rate; and $j_{z}$ is the current density.

The simulations were performed using the micromagnetic simulation package MuMax3 [52] at zero temperature (T = 0 K). We considered a hybrid ferromagnet/synthetic antiferromagnet (FM/SAF) structure. To realize the theoretical model, we chose Pt/CoNi/Ru/Co/Pt multilayered heterostructures to create the skyrmion mirroring effect. The saturation magnetization of CoNi ranged from 0.49 × 10^6^–1.1× 10^6^ A/m and the saturation magnetization of Co ranged from 0.2 × 10^6^ 1.35 × 10^6^ A/m. To simplify our model, the saturation magnetization was set to M_sat_ = 1.2 × 10^6^ A/m. The size of the unit cell was 1 nm × 1 nm × 1 nm.

## 3. Results and Discussion

### 3.1. Skyrmion Stability

First of all, we show the impact of the inter-modular antiferromagnetic coupling *A*_inter_ on the stability of skyrmions at the different intra-modular DMI coupling strengths for the base module (*D*_B_) when the DMI strength for the top module was fixed at *D* = 2.0 × 10^−3^ J/m^2^. In the SAF section, with vanishing inter-modular coupling (*A*_inter_ = 0), the two modules were decoupled completely, and each module was isolated. In this case, the skyrmion stabilized at *D*_B_ = 1.6–2.3 mJ/m^2^ with only *D*_B_ = 2.0 mJ/m^2^ (diameter of skyrmion d = 36 nm) shown in Figure 1c,d. An increase in *A*_inter_ results in an enhanced coupling between the top and bottom modules. The range of *D*_B_ that supports stable skyrmions expanded to 1.4–2.4 mJ/m^2^, which corresponded to the variations in the skyrmionic diameter, which ranged from 22 to 54 nm. When *A*_inter_ was above 0.5 × 10^−13^ J/m, the range was further extended to 1.2–2.4 mJ/m^2^, and the diameter varied from 18 nm to 54 nm. A larger *A*_inter_ led to a smaller skyrmion diameter and increased the ferromagnetic surroundings compared to an isolated one, resulting in improved stabilization of the skyrmion. In addition, the stabilization of skyrmions in the SAF section also depended on the threshold value $D_{c}=4\sqrt{AK}/\pi$ ≅ 1.2 mJ/m^2^ required to create a skyrmion. Although the inter-modular coupling lowered the energy, the skyrmion in the lower module did not exist, since the *D*_B_ fell below the threshold value *D*_c_. In the opposite limit, when the DMI value of the bottom modular was 2.8 mJ/m^2^, the magnetic structure was a helical state.

### 3.2. Skyrmion Mirroring Effect

To drive the skyrmion motion, we turned to STT via a charge current applied along the x direction. We assumed that, in the SAF section, the current flows in both the top and base modules and a skyrmion was first placed on the left boundary of the FM section close to the SAF section. In Figure 2a–c, we show the trajectory of the skyrmion under different current densities. Under a rather low current density, (*J*_x_ = 1.0 × 10^10^ A/m^2^), the skyrmion was blocked at the interface between the SAF and the FM sections, as shown in Figure 2a. The skyrmion moved with a velocity of 0.04 m/s along the interface due to the finite y-component of the velocity induced by the Magnus force until it was annihilated at the edge. The energy barrier between the SAF and FM sections contributed to the blocking of the skyrmion [53]. Due to such blocking, the total energy (of the whole structure) and the topological charge of the skyrmion almost remained constant in time, as shown in Figure 2d,e. With a higher current, the skyrmion entered the SAF section but was still pinned at the interface, as shown in Figure 2b. Accompanying such “trespassing” is a sudden energy drop, as shown in Figure 2d, indicating that such crossing must overcome an energy barrier. The further decrease in the topological charge in Figure 2e shows that the energy variation is related to the topological property (i.e., the energy barrier is associated with the topological transition of the spin textures). Interestingly, the ferromagnetic skyrmion could enter the SAF section and transit to a skyrmion pair when the current was higher than the value *J*_x_ = 2.5 × 10^12^ A/m^2^, as shown in Figure 2c. By analyzing the energy profile in Figure 2d, it was found that besides the first peak for *J*_x_ = 1.2 × 10^12^ A/m^2^, there was another peak when the entire skyrmion crossed the interfacial edge. For the former, the left part of the skyrmion entered the interfacial edge by the contraction in skyrmion size. In contrast, for the second peak, the right part of the skyrmion passed the interfacial edge through the expansion in skyrmion size. In order to pass the interfacial easily, the spins in the vicinity of skyrmion edge should be closer to a ferromagnetic state. For the second peak, the helical period was enlarged under the effect of current and pinning at the interfacial edge. In this case, the skyrmion was in a metastable state and the total energy was raised to the maximum due to the deformation. When the skyrmion bubble passed the interfacial, the skyrmion configuration recovered and total energy of the system was lowered. This led to a large energy drop when the time varied from 1 ns to 1.5 ns, as shown in Figure 2c. During this process, a skyrmion in the bottom module remained, and a skyrmion bubble (a skyrmion at breathing mode) in the top module transitioned to a skyrmion, which contributed to an opposite topological charge compared with that in the bottom module. This resulted in the total topological charge decreasing to zero from 0.1 in Figure 2e, and it further corroborated that there was also an energy barrier related to the transition between the skyrmion and the skyrmion bubble in the top module. Similarly, when a weak current (such as 1.2 × 10^12^ A/m^2^) was directed along the negative x axis, the antiferromagnetic coupled skyrmion was blocked at the interface between the SAF and the FM sections due to the energy barrier. With the current increasing to 2.5 × 10^12^ A/m^2^, the skyrmion at the top module was annihilated and only the skyrmion at the bottom module passed the interfacial and moved to the FM region.

In order to study the effect of the energy barrier on skyrmion duplication, we further plotted the temporal evolution of the local energy density when the current density was above the threshold value of the skyrmion passing the interfacial edge in Figure 3a–e. For an initially isolated skyrmion, the energy decayed from the edge to the center, as shown in Figure 3a. When the current was injected, the left edge of the skyrmion was blocked, as shown in Figure 3b. The interaction of the energy barrier and the spin torque led to squeezing of the skyrmion diameter, and the energy density of the skyrmion was increased in the single layer part compared with Figure 3a. In contrast, the skyrmion expanded when the right edge of the skyrmion was pinned, as shown in Figure 3c,d. The deformation of the skyrmion shape made it easier for the skyrmion to cross the interfacial edge. The energy barrier was 0.132 eV for the interlayer exchange coupling *A*_inter_ = 5 × 10^−13^ J/m and current density J = 2× 10^12^ A/m^2^, and the energy barrier was 0.156 eV *A*_inter_ = 1 × 10^−13^ J/m and the current density J = 2.4 × 10^12^ A/m^2^, as shown in Figure 3f. To efficiently drive the skyrmion motion, we compared the effects on skyrmion duplication when STT and SOT were separately applied to drive the skyrmion. We plotted the total energy evolution of skyrmion motion driven by SOT in Figure 3h,i. For a lower interlayer exchange interaction, *A*_inter_ = −5 × 10^13^ J/m, the threshold value of the current was 1.6 × 10^11^ A/m^2^ in Figure 3h. When the interlayer exchange interaction increased to *A*_inter_ = −1 × 10^12^ J/m, the threshold value of the current was enhanced to 2.0 × 10^11^ A/m^2^, as shown in Figure 3i. The trend is further shown in Figure 3j. The threshold value of the current first increased with the interlayer exchange coupling and then went to saturation. In this case, the spin torque together with interlayer coupling was sufficient to tilt spins and create skyrmions. The increased interlayer exchange coupling only retained the spin configurations of skyrmions. Hence, the threshold value of the current was saturated. In contrast, under the effect of STT, the STT in the mirror layer also stabilized the ferromagnetic background, leading to the increase in the threshold value of the current.

### 3.3. The Separation of the Antiferromagnetically Coupled Skyrmions

We further considered the influence of interlayer exchange coupling on skyrmion motion in Figure 4. When the current is injected into the system, two possible mechanisms contribute to the separation of the coupled skyrmions. One is related to the competition between repulsive forces because of the skyrmion Hall effect and the interlayer exchange coupling. When the current is relatively weak, the coupled skyrmions driven by SOT will move together, as shown in Figure 4a. In contrast, if the current is sufficiently strong, the coupling between bilayer skyrmions is weakened due to the increasing skyrmion Hall effect, and the skyrmions would be separated as shown in Figure 4b, which is consistent with previous results [31].

The dependence of skyrmion separation on the current density and interlayer coupling is further summarized in Figure 4c. When the interlayer coupling was above 6 × 10^−14^ J/m, there was no skyrmion separation such as when the spacing layer was chosen to be Ir. The other mechanism was related to the variation in the thickness of the spacing layer. For a wedge-shaped spacing layer, the interlayer roughly decreased with increasing thickness. To mimic the variation, we studied the skyrmion motion in a nanostrip with a gradient in the interlayer exchange coupling among the different subregions in Figure 4d. As expected, the antiferromagnetic coupled skyrmions were separated when they were driven to the region of weak interlayer exchange coupling in this system.

### 3.4. Skyrmion-Based Devices

Finally, we designed skyrmion-based devices as shown in Figure 5a–c. In Figure 5a, a wedged hybrid FM/SAF structure was used to generate skyrmion in the SAF region. The interlayer exchange coupling varied from 0–2× 10^−12^ J/m in the calculated results, which was the same order of magnitude as in the experiments (0–1.3 × 10^−12^ J/m) [38,44]. The possible candidate may be a Pt/Co/Ru multilayered structure and the spacing layer Ru ranged from 7 Å to 12 Å. The surface roughness of the materials may lower the contrast of interlayer exchange coupling. Nevertheless, the interlayer exchange coupling can be further enhanced by lowering the temperature and using the capping layer such as MgO on the synthetic antiferromagnetic structures given that the skyrmion is located at the left corner, which could be created in a circular defect by an electric current [54], and it is driven to the antiferromagnetic coupling region from the ferromagnetic region. When the skyrmion crosses the interfacial and energy barrier in the hybrid FM/SAF region, the passing or blocking of skyrmion in the SAF region can be controlled exactly. In Figure 5b, an extended hybrid FM/SAF structure was used to control the transport of the skyrmion line such as in skyrmion racetrack memory and correcting the error due to the skyrmion depinning or the annihilation of skyrmion. The left part was used to write a line of skyrmions and the right part was used to control the passing or blocking of skyrmion by a different current density to correct the error. A T-shaped top layer with an initial skyrmion for the right part could be used to add additional skyrmions that may be missed in the skyrmion line due to the pinning or other effects. Finally, we proposed a skyrmion complementor in the skyrmion-based racetrack memory. Given that the skyrmion is located at the left corner, it was driven to the antiferromagnetic coupling region from the ferromagnetic region. Then, a pair of skyrmions was created in region II and was separated in region III in Figure 5c. The skyrmion motion was impeded by the pinning effects due to the disorder or the presence of defects. This may modify the dynamics of the driven skyrmion and raise the threshold of the current density to drive the skyrmion. In addition, the interaction between the skyrmions and edges may affect the spacing between skyrmions in a skyrmion line. Hence, the current densities were different for crossing the interfacial edges and creating a skyrmion line, respectively. In this way, the number of skyrmions can be increased by repeatedly driving the skrymion from the left to right part. Note that the efficiency of the devices could be further enhanced by using a pair of antiferromagnetic skyrmions to duplicate skyrmions. The effect of decoupling of skyrmions was suppressed during the process of skyrmion transfer. The size of the top ferromagnetic layer should also be optimized to further suppress the effect of the decoupling of the skyrmion.

## 4. Conclusions

In summary, we studied the current-driven skyrmion motion and duplication in a hybrid FM/SAF structure. By manipulating the current, the ferromagnetic skyrmion can be switched to bilayer skyrmions, and the skyrmion Hall effect can be suppressed. Through a specific geometric design, the antiferromagnetically coupled bilayer skyrmions can be separated. This structure provides a method to create Néel skyrmions and an artificial Néel skyrmion lattice with a high efficiency for skyrmion -devices.

## Figures and Tables

**Figure 1 nanomaterials-13-00859-f001:**
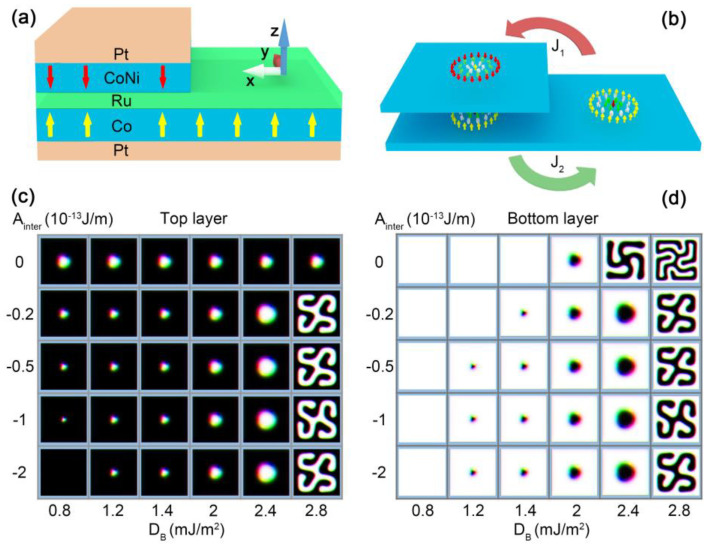
Schematic illustrations of the proposed structure. (**a**) Steplike multilayer tandem structure. Two ferromagnetic multilayers are separated by a spacing layer. Each ferromagnetic structure may require additional heavy metal (HM) layer(s) to create the DMI. (**b**) Schematic view of the system in the ferromagnetic modules. Across the SAF/FM boundary, an isolated ferromagnetic skyrmion can be switched to an antiferromagnetic coupled skyrmion pair and vice versa. Phase diagrams of the SAF structure for different inter-modular exchange coupling and intra-modular DMIs are shown for (**c**) the top module and (**d**) the base module. The size of the bilayer SAF structure was set to 300 nm × 300 nm × 2 nm and only a 100 nm × 100 nm region was shown to zoom in on the skyrmions. The Heisenberg exchange interaction of the intralayer was *A*_intra_ = 1 × 10 ^−11^ J/m. The perpendicular magnetic anisotropy in both layers was set to *K*_u_ = 3.2 × 10^5^ J/m^3^.

**Figure 2 nanomaterials-13-00859-f002:**
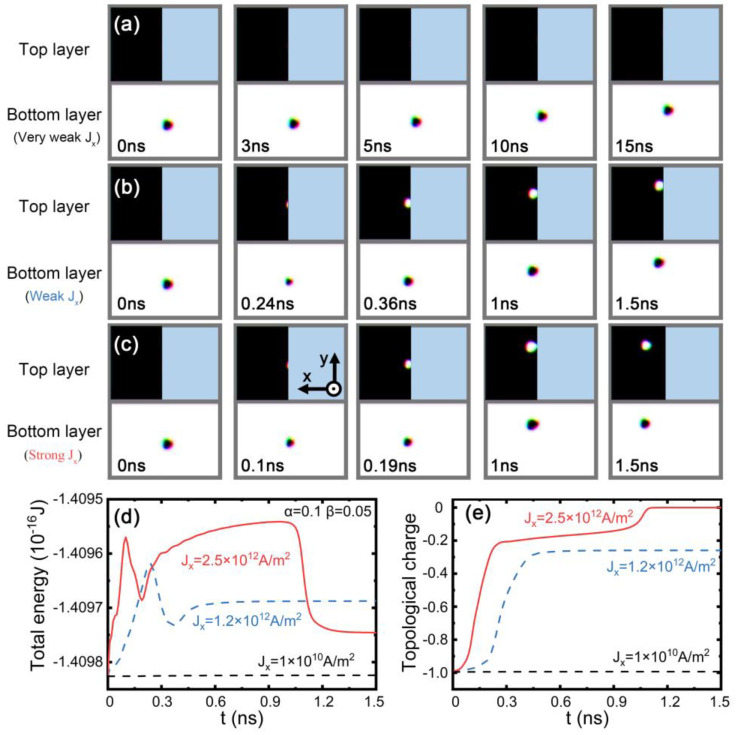
Time-resolved evolution of the motion of skyrmions for the top and bottom modules at different spin current densities. (**a**) 1.0 × 10^10^ A/m^2^, (**b**) 1.2 × 10^12^ A/m^2^, and (**c**) 2.5 × 10^12^ A/m^2^ along the x direction when the damping factor α = 0.1 and the nonadiabatic factor β = 0.05. Black and white regions indicate the top or bottom FM layer, respectively. The blue region denotes the absence of material. The interlayer antiferromagnetic coupling is −1 × 10^−12^ J/m. Temporal evolution of (**d**) the total energy and (**e**) the topological charge of the hybrid FM/SAF structure. The blue dashed line and red solid line indicate that the driving current density was 1.2 × 10^12^ A/m^2^ and 2.5 × 10^12^ A/m^2^, respectively. The size of the top ferromagnetic module and the bottom ferromagnetic module were set to 140 nm × 200 nm × 1 nm and 300 nm × 200 nm × 1 nm. For the top module, the parameters were *A*_intra_ =2 × 10^−12^ J/m, *K*_u_ = 3.2 × 10^5^ J/m^3^, and *D* = 1.2 mJ/m^2^; for the bottom module, the parameters were *A*_intra_ = 10 × 10^−12^ J/m, *K*_u_ = 3.2 × 10^5^ J/m^3^, and *D* = 2 mJ/m^2^.

**Figure 3 nanomaterials-13-00859-f003:**
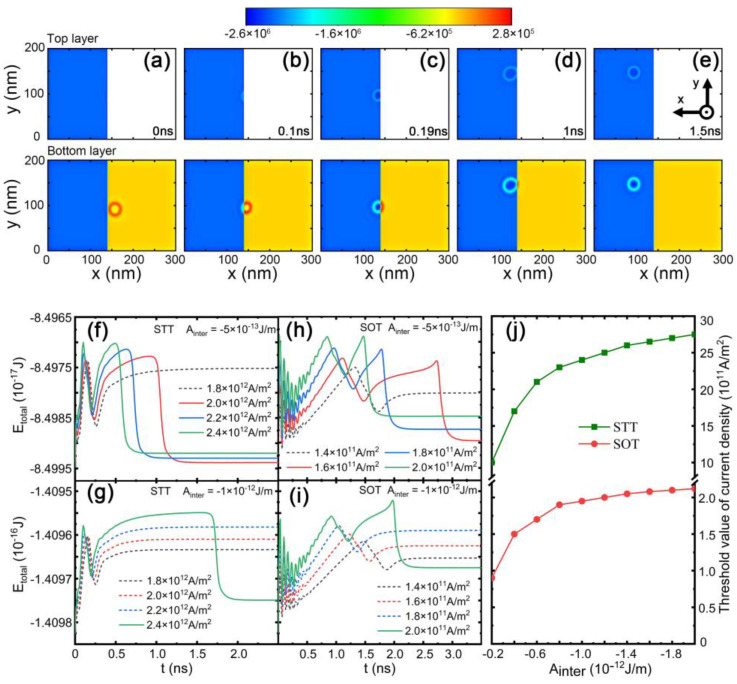
(**a**–**e**) The temporal evolution of the local energy density for the current density *J*_x_ = 2.4 × 10^12^ A/m^2^ under the effect of STT. The temporal evolution of total energy under different current densities: (**f**,**g**) for STT, (**h**,**i**) for SOT. The dashed lines indicate that the current was below the threshold value for a skyrmion to cross the interfacial edge. (**j**) The threshold value of the current density required to cross the interfacial edge as a function of interlayer exchange coupling.

**Figure 4 nanomaterials-13-00859-f004:**
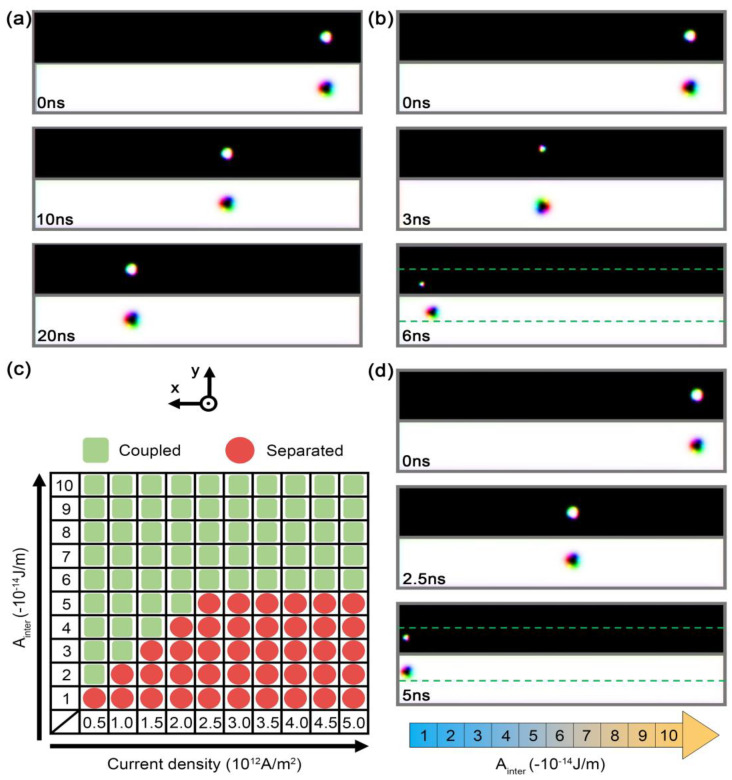
Snapshots of the motion of skyrmions with *A*_inter_ = −3 × 10^−14^ J/m for the current density (**a**) 5 × 10^11^ A/m^2^ and (**b**) 1.5 × 10^12^ A/m^2^. (**c**) The motion of an AFM coupled skyrmion for various interlayer exchange couplings and current densities. (**d**) Temporal evolution of the skyrmion motion in a nanostrip with an inhomogeneous gradient of the interlayer exchange interaction. The current was injected into the two layers along the –*x* direction and the strength was 2.5 × 10^12^ A/m^2^. The size of the SAF nanostrip was 400 nm × 60 nm × 2 nm. The parameters were *A*_intra_ = 2 × 10^−12^ J/m, *K*_u_ = 6 × 10^5^ J/m^3^, and *D* = 1.2 mJ/m^2^ for the top layer, and *A*_intra_ = 10 × 10^−12^ J/m, *K*_u_ = 3.2 × 10^5^ J/m^3^, and D = 2 mJ/m^2^ for the bottom layer, respectively.

**Figure 5 nanomaterials-13-00859-f005:**
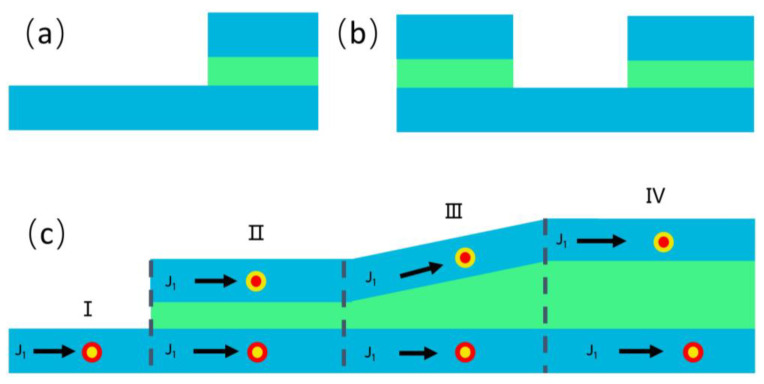
The wedged hybrid FM/SAF structures. (**a**) Skyrmion generator. (**b**) Skyrmion recoding. (**c**) Skyrmion complementor. The device includes reproduction, separation, and encoding. An initial skyrmion in region I was driven to the duplicating region, separation region, and encoding region.

## Data Availability

The data presented in this study are available on request from the corresponding author.

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
