# Peer review of "Mirroring Skyrmions in Synthetic Antiferromagnets via Modular Design"

_nanomaterials, 2023, doi:10.3390/nano13050859_

Round 1
Reviewer 1 Report
The topic of the paper is related to skyrmions in synthetic antiferromagnets (SAF) nanostructures with modular design. The interlayer exchange RKKY coupling (IEC) is exploited to pattern skyrmionic textures in hybrid SAF structures. By tuning the IEC, the mirrored skyrmionic structures can be separated, after being initially displaced by spin torque in some chosen locations. The current work proides an interesting theoretical prospective for possible further experiments in patterned spintronic devices. Having in view the interest of the skyrmionic applications in next generation spintronic devices for ICT applications, the originality and quality of the results I consider that the paper worth to be published.
Prior to acceptance, I would suggest the authors to address the following issues:
The stability of coupled skyrmions in SAF is driven by the competition between the Skyrmionic Hall effect-SHE- (classic equivalent “Magnus” effect) and the IEC. As shown and discussed, the two contributions can be tuned by the current density (SHE) and intensity of the IEC (controlled by the nature and the thickness of the non-magnetic spacer of the SAF. A larger IEC (case of Ir compared to Ru) would stabilize the AF mirroring and prevent the mirrored skyrmions separation. A gradient of IEC (A_inter) was proposed for the simulations illustrated in Fig.4 and the “device” illustrated in Fig.5. However, for future experimental implementation of the concept, I would suggest to “locate” the theoretical parameters within the experimental reality (adding relevant references):
- e.g. the values of the IEC (Ainter) compared with available experimental data, for both Ru and Ir spacers in perpendicular SAF systems.
- Based on experimental IEC vs t_spacer data, which experimental variation of thickness would be needed to get the A_inter gradient used in simulations (Fig. 4) and the II-IV zones in the wedged hybrid FM/SAF structure (Fig. 5) describing the proposed device? How does the experimental roughness would influence this concept ? Indicate some other issues can be experimentally used to enhance the IEC contrast between the zones?
Technical issues:
1/ Carefully check uniformity of notation in the text (and figure captions):
- write Ainter with “inter” as subscript, everywhere in the paper. In some places (e.g. line 120, 128, caption of figure 2,… the subscript is missing).
- concerning the units, J/m2 again “2” has to be a superscript (everywhere in the paper)- e.g. line 122.
2/ All the figures look “blurred”: please re-insert figures with better resolution, details (text, notations) are almost impossible to be read in the current figures.
Reviewer 2 Report
The key aspects of the study and some of the results have been reported by other groups, although not in one bite, e.g. AF coupled skyrmions can suppress Hall drift effect, production of skyrmions and reading of skyrmions, AFM skyrmions formed in synthetic AFM have recently been published but not mentioned (C.T. Ma et al, Scientific Report 2022), etc. etc.. The spin-orbit torque driven skyrmion motion in the modeled AFM, on the other hand, seems to be new. Overall, the skyrmion dynamics is computed in more detail than earlier work.
A comparison with experimental current density for skyrmion creation should be given.
In the discussion of driving skyrmions (figure 5), the pinning and interaction with edges must be considered.
The manuscript could be reconsidered for publication.
Reviewer 3 Report
Dear Editor,
The manuscript “Mirroring skyrmions in synthetic antiferromagnets via modular design” by Panluo Deng, Fengjun Zhuo, Hang Li and Zhenxiang Cheng simulates behaviour of magnetic topological textures (skyrmions) in metallic multilayer structure with antiferromagnetically coupled ferromagnetic films (SAF). Authors showed that a pair of skyrmions with opposite topological charge can be created by pushing a single skyrmion into an artificial antiferromagnet from a FM layer using a spin-polarized electric current. Such a skyrmion pair does not have skyrmion Hall effect. Lately, the skyrmion pair can be separated. Creating a skyrmion in a SAF is not trivial, since the SAF structure does not interact with external field. So, the idea of creation of a skyrmion pair in SAF by pushing a skyrmion from a FM layer looks interesting (note that recently similar approach was used to create a bimeron). Therefore, I would recommend the paper for the publication but only after the authors make certain improvements and address all concerns shown below.
1. I believe that the abstract section is slightly misleading. Authors claim that they can create a skyrmion using RKKY interaction. In fact, they just transform existing skyrmion (created by some other mean) to a pair of skyrmions at the boundary between FM and SAF layers. Moreover, they need to apply a current to realize this transformation.
2. The motivation part is not clear for me. Authors claim that their approach allow to create a skyrmion “exactly” and deliver it accurately to a desired place. However, their “device” needs an existing skyrmion to work. So, a skyrmion should be created by some other mean, placed into their device and then the device creates the second skyrmion. Why don’t you just use the initial skyrmion? I would instead just say that the paper proposes a method of creating a skyrmion pair in a SAF which is not trivial. So, the method proposed in the manuscript is valuable.
3. The expression for system energy used in simulations should be shown in the text.
4. Is there magneto-crystalline anisotropy in the system?
5. Is there magneto-dipole interaction in the system?
6. Is the interlayer exchange coupling local?
7. Fully metallic system is considered in the paper. So, we have few metallic layers one on top of each other and there is no insulating spacer between them. Therefore, I do not understand what authors mean when they say “we apply current to the bottom layer”. The current should flow along all layers in metallic systems.
8. Why at A_inter=0 skyrmion exists in bottom layer, but does not exist in the top layer? What is the difference between the layers in term of material parameters?
9. What is the magnetic anisotropy of the layers used to obtain Fig. 1?
10. Can authors explain why the final energy of the system (see Fig. 3) after skyrmions enters SAF region is different depending of the spin-torque mechanism (SOV Vs STT)? The current itself does not contribute to the system energy.
11. Why at A_inter>6*10^(-14) J/m, one cannot decouple the skyrmions whatever the current is?
12. Can authors elaborate more on how the device shown in Fig. 5(b) works. Single sentence in not enough I believe.
Style:
1. Figure quality (resolution) must be improved
2. Sometime, author use just lower case in notations (such as Ainter) and sometime authors uses subscripts (like ). Single style should be chosen. Usually, people use subscripts rater than lower case letter.
3. I would add the material parameters to the text (not to the figure captions).
Reviewer 4 Report
The manuscript by P. Deng et al. presents a solid computational study of the formation of mirror skyrmions in the magnetic heterostructures. The results are clearly presented, they will be with no doubt very interesting for the readers of Nanomaterials. I propose only minor corrections to improve the presentation.
1) Quality of figures: please, improve the resolution of figures as high as possible within the journal's guidelines.
2) Please, introduce the SOT abbreviature in the text.
3) Figure 1: please, explain the symbol and the magnetic structure for Db=2.8.
4) The calculations yield the energy barrier for the formation of mirror skyrmions as shown in Figure 3f. Please, provide a typical values of this energy barrier in electron volts eV, because it characterizes the energy bonding between two skyrmions with opposite topo charges.
Round 2
Reviewer 2 Report
publish.
Reviewer 3 Report
Dear Editor,
authors addressed my questions and comments in a reasonable manner. They made necessary manuscript improvements as well. I think, now the manuscript is ready for publications.